# A multicenter study investigating the genetic analysis of childhood steroid-resistant nephrotic syndrome: Variants in *COL4A5* may not be coincidental

**Sheng Li[1,2], Miaoyue Hu[1], Chao He[1,2], Yu Sun[1], Weifang Huang[1], Fengying Lei[1], Yunguang Liu[3], Zengpo Huang[3], Yongqiu Meng[4], Wenjing Liu[4], Xianqiang Lei[5], Yanfang Dong[5], Zihui Lin[6], Chunlin Huang[6], Rihong Zhao[7], Yuanhan Qin [1]\***

1 Department of Pediatrics, The First Hospital of Guangxi Medical University, Nanning, China, 2 Department of Pediatrics, The First Affiliated Hospital of University of South China, Hengyang, China, 3 Department of Pediatrics, Affiliated Hospital of Youjiang Medical College for Nationalities, Baise, Guangxi Province, China, 4 Department of Pediatrics, Guigang People's Hospital, Guigang, China, 5 Department of Pediatrics, Liuzhou Maternity and Child Healthcare Hospital, Liuzhou, China, 6 Department of Pediatrics, Maternity and Child Healthcare of Guangxi Zhuang Autonomous Region, Nanning, China, 7 Department of Pediatrics, Affiliated Hospital of Guilin Medical University, Guilin, China

\* qinyuanhan2022@163.com

**Data Availability Statement:** All relevant data are within the manuscript and its Supporting Information files.

**Funding:** This research was funded by Innovation Project of Guangxi Graduate Education (NO:

## Abstract

This study aimed to discuss the pathogenic hereditary factors of children with steroid-resistant nephrotic syndrome (SRNS) in Guangxi, China. We recruited 89 patients with SRNS or infantile NS from five major pediatric nephrology centers in Guangxi, and conducted a retrospective analysis of clinical data. Whole-exome sequencing analysis was also performed on all patients. The risk of progression to chronic kidney disease (CKD) was assessed using the Kaplan-Meier method and Cox proportional hazards model. The study included 69 male and 20 female participants from 86 distinct families, with the median age of disease onset being 48 months (interquartile range: 24–93). Overall, 24.7% had a family history of SRNS, whereas 13.5% exhibited extra-kidney manifestations. We identified disease-causing variants in 24.7% (22/89) of patients across eight screened genes. The most frequently detected variant was found in *COL4A5*, followed by *NPHS2* (5.6%), *NPHS1* (2.2%), *PAX2* (2.2%), *WT1* (1.1%), *LMX1B* (1.1%), *NUP105* (1.1%), and *COL4A6* (1.1%). Twelve of the 26 pathogenic variants were determined to be de novo. Based on gene detection results, pathogenic variants were categorized into two groups: identified and unidentified variants. The identified variant group demonstrated a significant association with positive family history, steroid resistant-style, and response to immune therapy (*P*<0.001). Patients with the identified genetic variant were approximately ten times more likely to develop CKD (*P*<0.001) than those in the unidentified group at the last follow-up. Kidney biopsy was performed on 66 patients, and minimal change disease was the most prevalent histopathological diagnosis (29 cases; 32.6%). These findings suggest that children diagnosed with SRNS exhibit a diverse range of genetic alterations. We identified the *COL4A5* variant as the predominant genetic abnormality and a low frequency of *NPHS1* gene involvement in

YCBZ2021047); Research Basic Ability Enhancement Project of Young and Middle-aged Teachers in Guangxi Universities(NO: 2021KY0096); Guangxi Health committee self-funded scientific research project (NO: Z20190155) and Guangxi Medical High-level Backbone Talent "139" Plan.

**Competing interests:** The authors have declared that no competing interests exist.

these children. Gene variants may serve as an independent predictor for SRNS progression to CKD.

## Introduction

Idiopathic nephrotic syndrome (INS) is one of the most common glomerular diseases observed in childhood, with an incidence ranging from approximately 2 to 6.5 per 100,000 children yearly, depending on racial and geographical differences [1,2]. INS is characterized by considerable proteinuria (40 mg/m$^2$ per hour or 24-h urinary protein excretion $\geq$50 mg/kg), resulting in hypoalbuminemia and subsequently causing edema and hyperlipidemia. Approximately 10–15% of INS cases do not achieve complete remission after 4–6 weeks of corticosteroid therapy, these cases are termed steroid-resistant NS (SRNS) [3]. Moreover, SRNS is more likely to progress to chronic kidney disease (CKD) [4]. Despite its unclear pathogenesis, up to 30% of children with SRNS may have underlying monogenic etiologies, which are often unresponsive to immunosuppression [5].

With the advent of next-generation sequencing, more than 60 genes linked to monogenic SRNS have been identified. Presently, whole exome sequencing (WES) has become an indispensable diagnostic tool for diagnosing and assessing SRNS. Furthermore, WES may provide a better basis for selecting different treatment options, predicting recurrence risk after kidney transplantation, and understanding the stratification between immunological and/or circulating factor and genetic diseases [6].

However, the high heterogeneity of genetic SRNS highlights the importance of considering different regions, ethnicities, and molecular diagnoses. Unlike the North American and Western European populations, where approximately a third of SRNS can be secondary to an underlying genetic etiology, with *NPHS2* variants being the most common [5,6], in China, Japan, and Africa, the most frequently mutated genes are *WT1*, *NPHS1*, and *APOL1*, respectively [7–9]. Therefore, genetic variability across ethnicities underscores the need for local data. In this report, we aimed to describe the genetic and clinical spectrums of 89 SRNS cases from five tertiary pediatric nephrology centers in Guangxi Zhuang Autonomous Region and identify the link between genetic variants and clinical predictability of disease progression.

## Materials and methods

### Ethics consideration

This study was approval by the Ethics committee for the Evaluation of Clinical Research Projects at the first affiliated Hospital of Guangxi Medical University (approval number 2023-E753-01). Informed consent was obtained from the patient or legal guardian. The participants signed a written informed consent for genetic studies of patients.

### Study participants

This study enrolled 89 patients with SRNS or infantile NS conducted at an international cohort of 86 families (ages between 1 and 216 months) at five major pediatric nephrology centers in Guangxi, China from January 1,2017 to October 31,2023. All the clinical data were collected from October 1,2023 to November 30,2023. Patients who had not experienced remission after receiving 4–6 weeks of 2 mg/kg of prednisolone (maximum dose of 60 mg/day) at the time of referral to our cohort were deemed as initial SRNS. Additionally, late SRNS was defined as

persistent proteinuria during $\geq$4 weeks of corticosteroids following one or more remissions [10,11]. Infantile NS was defined as SRNS with an onset age of $\leq$12 months. Patients with SRNS are defined as "immunosuppressant resistant" in the absence of complete remission after 12 months (calcineurin inhibitors [CNIs] for over 6 months) of treatment with two mechanistically distinct steroid-sparing agents administered at standard doses [3]. Exclusion criteria included participants with secondary glomerulonephritis, such as Henoch-Schönlein purpura nephritis, membranous nephropathy, immunoglobulin A nephropathy, lupus nephritis, diabetic nephropathy, and obesity-related glomerulopathy [12].

Demographic and clinical data of 89 patients with SRNS were collected and analyzed from the patient's medical records. These data included the age of onset, sex, ethnicity, family history, extra-kidney manifestations, blood pressure, routine urine, 24-h urinary protein concentration, serum creatinine (last follow up), kidney histology, response to immunosuppressants, kidney outcome at follow-up, and other laboratory tests. Hypertension was defined as exceeding the 95th percentile for systolic or diastolic blood pressure for age, sex, and height [13]. Estimated glomerular filtration rate (eGFR) was calculated using the original Schwartz formula: k × body length (cm)/serum creatinine level (mg/dL) [14,15]. Normal eGFR was defined as eGFR$\geq$90 mL/min/1.73 m$^2$.CKD was staged based on published CKD classification [16,17]. The primary endpoint for kidney outcome evaluation included major morbidity events, such as progression to CKD or mortality caused by kidney disease.

## Genetic analysis

WES was performed at MyGenostics Gene Technologies (Beijing, China) using genomic deoxyribonucleic acid isolated from the whole blood of the patient and their family members (if available). Segregation of identified variants was verified via polymerase chain reaction and Sanger sequencing in parents unless trio exome analysis had been performed. Variants were classified based on the American College of Medical Genetics and Genomics (ACMG) guidelines [18]. Based on the results of genetic testing, participants were divided into two groups: identified (existence of pathogenic gene variant) and unidentified group (without pathogenic gene variant).

## Statistical analysis

Continuous variables are presented as medians and interquartile ranges (IQR), and categorical variables as counts and proportions. Categorical variables were analyzed using the chi-squared or Fisher's exact test. The Mann–Whitney U test was used to compare median differences between two continuous variable groups.

Results are reported as hazard ratios (HR) with 95% confidence intervals (CI) and represented in Kaplan–Meier curves. Univariate and multivariate Cox regression analyses were performed to calculate HR and 95%CI after controlling for potential confounders. We included interaction terms between each specified subgroup in Cox proportional hazards models adjusted for age, family history, extra-kidney manifestations, and hypertension to calculate $P$-values for interaction. Statistical significance was set at a 2-tailed $P$<0.05. All statistical analyses and diagrams are presented in R software (version 4.3.1).

## Results

### Clinical features

In total, 89 patients with SRNS (male: female = 69:20) from 86 different families were included in this study. The median age at disease onset was 48 months (IQR 24,93). No consanguineous

marriages were reported in our cohort. In 22 patients who had positive family histories of proteinuria, NS, or CKD (**Table 1**), extra-kidney manifestations at presentation were reported in 12 (13.5%) cases, with neurological defects being the most frequent (41.7%; **S1 Table**).

**Table 1. Clinical features of 89 children with SRNS.**

| Variables | Total (n = 89) | Unidentified (n = 67) | Identified (n = 22) | P-value |
|---|---|---|---|---|
| Age of onset, Median (Q1,Q3) | 48.00 (24.00, 93.00) | 41.00 (19.50, 85.50) | 76.00 (48.00, 109.75) | **0.027** |
| Sex, n (%) | | | | 0.248 |
| Male | 69 (77.5) | 54 (80.6) | 15 (68.2) | |
| Female | 20 (22.5) | 13 (19.4) | 7 (31.8) | |
| Nation, n (%) | | | | 0.436 |
| Han | 53 (59.6) | 40 (59.7) | 13 (59.1) | |
| Zhuang | 32 (36.0) | 25 (37.3) | 7 (31.8) | |
| Others | 4 (4.5) | 2 (3.0) | 2 (9.1) | |
| Family history, n (%) | | | | < **0.001** |
| no | 67 (75.3) | 57 (85.1) | 10 (45.5) | |
| yes | 22 (24.7) | 10 (14.9) | 12 (54.5) | |
| Extra-kidney manifestations, n (%) | | | | 0.481 |
| no | 77 (86.5) | 59 (88.1) | 18 (81.8) | |
| yes | 12 (13.5) | 8 (11.9) | 4 (18.2) | |
| Steroid resistant-style, n (%) | | | | < **0.001** |
| not used | 5 (5.6) | 1 (1.5) | 4 (18.2) | 0.012 |
| initial-resistant | 66 (74.2) | 48 (71.6) | 18 (81.8) | 0.412 |
| late-resistant | 18 (20.2) | 18 (26.9) | 0 (0) | **0.006** |
| 24hUpro, Median (Q1,Q3) | 2604.14 (1442.00, 3972.80) | 2795.80 (1713.25, 4459.31) | 2336.65 (1096.41, 3488.52) | 0.126 |
| Scr, Median (Q1,Q3) | 36.00 (25.00, 58.00) | 32.00 (24.50, 53.50) | 51.50 (29.50, 114.75) | **0.014** |
| eGFR, Median (Q1,Q3) | 113.31 (84.98, 167.52) | 124.98 (97.34, 167.00) | 92.01 (37.34, 146.46) | 0.071 |
| Hypertension, n (%) | | | | 0.065 |
| No | 77 (86.5) | 61 (91.0) | 16 (72.7) | |
| yes | 12 (13.5) | 6 (9.0) | 6 (27.3) | |
| kidney histology, n (%) | | | | **0.021** |
| not done | 33 (37.1) | 30 (44.8) | 3 (13.6) | 0.009 |
| MCD | 29 (32.6) | 19 (28.4) | 10 (45.5) | 0.138 |
| FSGS | 18 (20.2) | 11 (16.4) | 7 (31.8) | 0.119 |
| MsPGN | 4 (4.5) | 4 (6.0) | 0 (0) | 0.568 |
| MN | 4 (4.5) | 3 (4.5) | 1 (4.5) | >0.99 |
| DMS | 1 (1.1) | 0 (0) | 1 (4.5) | 0.247 |
| Response to immune therapy, n (%) | | | | < **0.001** |
| not used | 13(14.6) | 7 (10.4) | 6 (27.3) | 0.079 |
| responder | 50 (56.2) | 48 (71.6) | 2 (9.1) | <**0.001** |
| no-responder | 26 (29.2) | 12 (17.9) | 154(63.6) | <**0.001** |
| Follow-up time, Median (Q1,Q3) | 39.00 (24.00, 65.00) | 46.00 (25.00, 67.50) | 33.00 (24.00, 55.50) | 0.122 |
| Prognosis, n (%) | | | | < **0.001** |
| normal eGFR | 71 (79.8) | 60 (89.6) | 11(50.0) | <**0.001** |
| CKD2-4 | 7(7.9) | 4 (6.0) | 3(13.6) | 0.358 |
| CKD5 | 9 (10.1) | 3 (4.5) | 6 (27.3) | **0.006** |
| mortality | 2 (2.2) | 0 (0) | 2 (9.1) | 0.059 |

Scr, Serum creatinine; eGFR, estimated glomerular filtration rate; FSGS, focal segmental glomerulosclerosis; MCD, Minimal change disease; MsPGN, Mesangial proliferative glomerulonephritis; MN, membranous nephropathy; DMS, Diffuse Mesangial sclerosis; CKD, chronic kidney disease.

Steroid therapy was administered to 84 patients, and all demonstrated steroid resistance (except five of infantile NS), consisting of 66 initial SRNS and 18 late SRNS. No gene variants were found in late SRNS. Among the 56 (62.9%) patients with SRNS who underwent kidney biopsy, the most common histopathological diagnosis was minimal change disease (MCD, 29 cases; 32.6%), followed by focal segmentary glomerulosclerosis (FSGS, 18 cases; 20.2%). Among the 89 patients, 76 patients received immune therapy, with 50 (56.2%) patients showing a response and 26 (29.2%) showing no response (**Table 1**).

## Detailed genetic results

Of the 89 cases sequenced, no sex difference was observed between cases with identified and unidentified genetic variants. The detection rate of disease-causing variants was 24.7% (22/89), with patient 62 having a combination of two pathogenic gene variants. Moreover, 26 (likely) pathogenic variants were identified in eight genes. Eight (34.8%) participants in the genetic variant group had autosomal recessive variants, whereas four (17.4%) and 11 (47.8%) had autosomal dominant and X-linked variants, respectively. Among the 26 variants, 12 (46.2%) variants were de novo (**Table 2**).

*COL4A5* was the most common causative gene (10, 11.2%), followed by *NPHS2* (5, 5.6%), *NPHS1* (2, 2.2%), *PAX2* (2, 2.2%), *WT1* (1, 1.1%), *LMX1B* (1, 1.1%), *NUP105* (1, 1.1%), and *COL4A6* (1, 1.1%) (**Fig 1A and 1B**). No disease-causing variants were detected in children in the first year of life. However, the rate of these variants increased with age: approximately 22.2% at age 12–60 months, 30.0% at age 60–108 months, 44.4% at age 108–144 months, and 25.0% after age 144 months (**Fig 2**).

## Genotype-phenotype correlations in SRNS or infantile NS

The identified variant group was significantly associated with a positive family history, steroid resistant-style, response to immune therapy ($P<0.001$), and onset age (76 versus 41 months, $P = 0.027$). Genetic abnormalities were observed in seven (37.4%) of the patients with FSGS and 45.5% of the patients with MCD. FSGS was found more frequently in patients with genetic variants than in those without genetic variants, accounting for only 16.4% (**Table 1**). Ten participants (eight male and two female) from nine families had *COL4A5* variants. Moreover, the median age of onset was 108.9 months (IQR 90.3, 130), ranging widely from 33 to 168 months, significantly higher than that of all participants. Seven of these patients had family histories of kidney disease, and all patients exhibited an X-linked dominant mode of inheritance of *COL4A5* variants. Eleven gene variants were discovered, including three splicing variants, four missense variants, and four frameshift variants. Six of these variants were de novo. Light microscopy revealed FSGS and MCD in five and three cases, respectively. Moreover, one patient did not undergo a kidney biopsy. Electron microscopy (EM) revealed diffuse foot process effacement in seven patients, with variable thinning of the glomerular basement membrane or lacerated changes. Using clinical features and genetic reports, only one of them had sensorineural deafness, and none had ocular abnormalities of the classical Alport syndrome (AS) phenotype. All patients exhibited resistance to conventional steroid therapy. The median duration of follow-up from onset was 31.1 months (IQR 15.8, 44.5). Three patients maintained normal eGFR, six progressed to CKD stages 2–4, and patient 221 eventually died. Of the 10 children with SRNS and *COL4A5* gene variants, nine were Han nationality, and one was Miao nationality (**Tables 2 and 3**).

In the genetic variant group, *NPHS2* variants were the second most frequently detected variants in five patients with a mean age at onset of 55.6 months. Three of them had family histories. Four of them developed into CKD during follow-up, and one died (**Table 3**). Patient 62,

**Table 2. Details of Pathogenic variants genes identified in 22 SRNS (22/89).**

| Patients/ID | Gene | Exon/Intron (zygosity) | Type of variant/ Reported or novel | Mode of inheritance | variant | Protein function prediction | Effect according to ACMG | Evidence of pathogenicity |
|---|---|---|---|---|---|---|---|---|
| 141 | NPHS1 | 19(hom) | Duplicate; Reported | AR | C.2633dup(p.Asn878fs) | D | Pathogenic | PVS1+PM2 |
| 276 | NPHS1 | exon24-29 (het) | Delete;De novo | AR | exon24-29del | - | Likely pathogenic | PVS1+PM2 |
| 14 | NPHS2 | 7(het) | Missense;Reported | AR | c871C>T(p.Arg291Trp) | D | Likely pathogenic | PM3+PS3+PM2+PP1+PP3 |
| 59 | NPHS2 | 4(het), 5(het) | Splicing,De novo; Missense,Reported | AR | c.534+2T>C(splicing); c.593A>C(p.E198A) | -,D | Pathogenic; Pathogenic | PVS1+PM2+PM3+PP4;PS1 +PM2+PM3+ PP3+PP4 |
| 62 | NPHS2; COL4A6 | 2(het), 4(het); 35(hemi) | Missense,Repoted; Delins,De novo; Missense,Reported | AR/XLR | c.370T>C(p.C124R) c.491_493del,AGG(p. K164_V165delinsI); c.3478 G>A (p.G1160R) | D;-;D | Likely pathogenic; Likely pathogenic; Likely pathogenic | PM1+PM2+PM5+PP3;PM2 +PM3+PM4;PM2+PM3 +PM4 |
| 268 | NPHS2 | 4(hom) | Frameshift, Reported | AR | c.467del(p.Leu156Cys fsTer25) | - | Pathogenic | PVS1+PM2+PM3 |
| 269 | NPHS2 | 4(hom) | Frameshift, Reported | AR | c.467del(p. Leu156CysfsTer25) | - | Pathogenic | PVS1+PM2+PM3 |
| 2 | COL4A5 | 43(het) | Splicing,De novo | XLD | c.3997+1G>T(splicing) c.1447+5G>A | - | Likely pathogenic | PVS1+PM2 |
| 61 | COL4A5 | 53(het) | Missense,Reported | XLD | c.G973A:p.G325R | LD | Pathogenic | PVS1+PS3+PM2 |
| 128 | COL4A5 | 32 (hemi) | Frameshift,De novo | XLD | c.2729delT(p. Leu910Trpfs*86) | D | Pathogenic | PVS1+PM2+PP4 |
| 129 | COL4A5 | 32 (hemi) | Frameshift,De novo | XLD | c.2729delT(p. Leu910Trpfs*86) | D | Pathogenic | PVS1+PM2+PP4 |
| 16 | COL4A5 | 18 (het) | Splicing,Reported | XLD | c.1032+3_1032 +6delAAGT | D | Likely pathogenic | PS4+PM2 |
| 43 | COL4A5 | 41 (het) | Missense,Reported | XLD | c.3764G>T(p.G1255V) | D | Likely pathogenic | PM1+PM2+PM5+PP3 |
| 107 | COL4A5 | 35 (hemi) | Frameshift,De novo | XLD | c.3041delC(p.P1014Lfs* 7) | - | Pathogenic | PVS1+PM2+PP4 |
| 105 | COL4A5 | 41 (hemi) | Splicing,De novo | XLD | C3605-1G>A | D | Pathogenic | PVS1+PS2+PM2 |
| 221 | COL4A5 | 25(het) | Frameshift,De novo | XLD | c.1A>G(p.Met1Val) | - | Likely pathogenic | PVS1+PM2 |
| 281 | COL4A5 | 32(hemi), 32(hemi) | Missense,Reported Missense, Reported, | XLD | c.2731G>A(p.Gly911Arg); c.2714G>A(p.Gly905Asp) | D | Pathogenic; Pathogenic | PS4+PM1+PM2+PM5+PP3; PS4+PM2+PP3 |
| 5 | WT1 | 9 (het) | Splicing,Reported | AD | c.1447+5G>A(splicing) | - | Pathogenic | PS2+PS4+PM2 |
| 17 | PAX2 | 2(het) | Missense,De novo | AD | c.122T>C(p.I41T) | D | Likely pathogenic | PS2+PM2+PP3 |
| 38 | PAX2 | 8(het) | Missense,Reported | AD | c.254G>T(p.Gly85Val) | D | Likely pathogenic | PVS1+PM2 |
| 133 | NUP107 | 11(het) | Nonsense,De novo | AR | c.935dup(p.Tyr312Ter) | - | Likely pathogenic | PVS1+PM2 |
| 87 | LMX1B | 5(het) | Missense,De novo | AD | c.794T>G(p.Val265Gly) | D | Pathogenic | PM1+PM2+PM5+PP3 |

with initial SRNS at 48 months old, had an elder sister who died of kidney disease at 12 years old. He was treated with CNIs; however, complete remission was not achieved. Two compound heterozygous pathogenic *NPHS2* variant (c.370T>C[p.C12R], c.491_493del AGG[p.

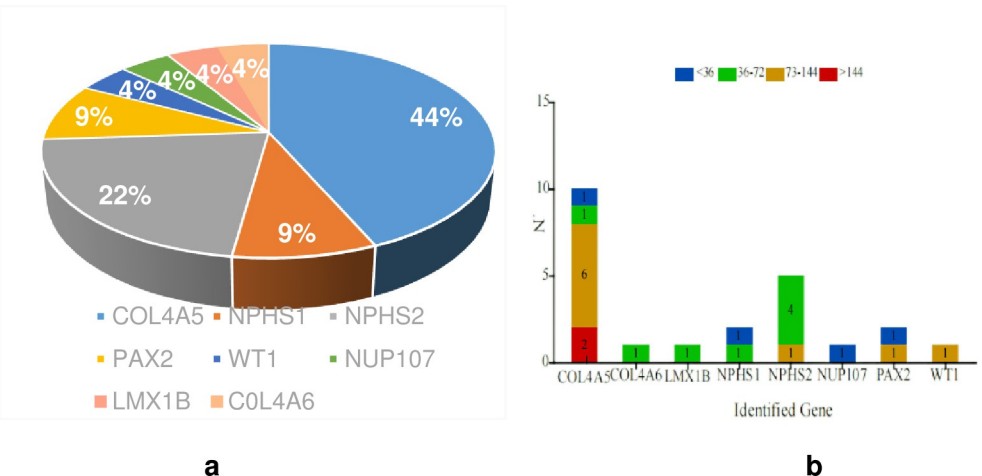

**Fig 1. Summary of identified genetic variants in the SRNS cohort.** (1a) Percentage of causative variant detected in this group. (1b) Distribution of genetic variants identified in different age groups.

K164_V165delinsI]) combined with hemizygote variant of *COL4A6* c.3478G>A (p.G1160R) was detected. A kidney biopsy revealed FSGS; his kidney function declined after a 2-year follow-up, and he died of CKD5 (**Table 2**). Patients 269 and 268, who are siblings, were diagnosed with initial SRNS at 48 and 53 months, respectively, achieving partial remission with immunosuppressants. Genetic testing indicated that both of them had a homozygous pathogenic variant on *NPHS2* c.467del (p.Leu156CysfsTer25), a zero-effect variation (frameshift variant), which may lead to the loss of function. Despite their kidney pathological findings revealing that both had FSGS, the outcomes considerably differed. Patient 269 progressed to CKD5 (follow-up of 58 months), whereas his younger brother was at the CKD2 stage (follow-up of 48 months) (**Tables 2 and 3**). *NPHS1* gene variants were uncommon in our study; however, they were observed in two patients. Neither of them had a family history or extra-kidney manifestations. One of them (patient 141) was unresponsive to immunosuppressants and

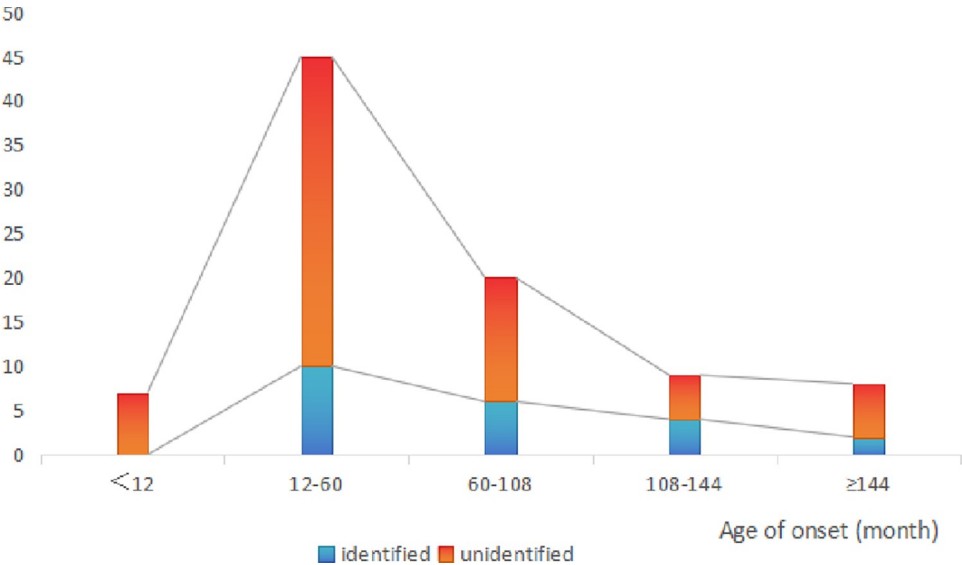

**Fig 2. Frequency map of gene variants detected in children with SRNS of different age groups.**

**Table 3. Genotypes and phenotypes of 22 SRNS patients with disease-causing variants.**

| Patients/ ID | Gene | Sex | Age at Onset (M) | Ethnicity | Family history | Extra-kidney manifestations | Response to immune therapy | Kidney Biopsy | Follow up time(M) | kidney outcome |
|---|---|---|---|---|---|---|---|---|---|---|
| 141 | NPHS1 | F | 15 | Zhuang | N | N | N | FSGS | 68 | CKD5 |
| 276 | NPHS1 | M | 43 | Zhuang | N | N | Y | ND | 4 | N |
| 14 | NPHS2 | M | 48 | Han | N | N | N | FSGS | 12 | N |
| 59 | NPHS2 | M | 81 | Zhuang | N | N | N | MCD | 24 | CKD5 |
| 62 | NPHS2; COL4A6 | M | 48 | Zhuang | Y | N | N | FSGS | 50 | DEAD |
| 268 | NPHS2 | M | 53 | Zhuang | Y | N | Y | FSGS | 48 | CKD2 |
| 269 | NPHS2 | F | 48 | Zhuang | Y | N | N | FSGS | 58 | CKD5 |
| 2 | COL4A5 | F | 168 | Han | Y | N | NU | ND | 32 | N |
| 61 | COL4A5 | M | 147 | Yao | Y | Sensorineural deafness | N | FSGS | 59 | CKD3 |
| 128 | COL4A5 | M | 115 | Han | Y | N | N | FSGS | 48 | CKD2 |
| 129 | COL4A5 | M | 115 | Han | Y | N | N | FSGS | 48 | CKD2 |
| 16 | COL4A5 | M | 124 | Han | Y | N | N | FSGS | 34 | CKD2 |
| 43 | COL4A5 | F | 33 | Han | N | N | N | FSGS | 34 | N |
| 107 | COL4A5 | M | 72 | Han | N | N | N | MCD | 30 | CKD2 |
| 105 | COL4A5 | M | 132 | Han | N | N | NU | MCD | 11 | CKD2 |
| 221 | COL4A5 | M | 94 | Han | Y | N | N | MN | 10 | DEAD |
| 281 | COL4A5 | M | 89 | Han | Y | N | NU | MCD | 5 | N |
| 5 | WT1 | F | 80 | Han | N | Mental-retardati-on;Atrial septal defect;46XY | NU | ND | 40 | CKD5 |
| 17 | PAX2 | F | 89 | Han | N | N | NU | FSGS | 27 | CKD3 |
| 38 | PAX2 | M | 33 | Han | N | Atrial septal defect;Prematue | N | ND | 34 | N |
| 133 | NUP107 | M | 27 | Zhuang | N | N | N | FSGS | 20 | CKD5,Kidney transplantation |
| 87 | LMX1B | F | 48 | Maonan | Y | Nail-patella defect,Atrial septal defect; | N | DMS | 98 | CKD5 |

AR:autosomal recessive; AD:autosomal dominant; XLD:X-linked dominant; XLR:X-linked recessive; ACMG:American College of Medical Genetics and Genomics Guidelines.

rapidly progressed to CKD5. The other patient exhibited partial remission with immunosuppressants and normal kidney function, with a follow-up period of only 4 months.

Other less common gene variants were identified in our study. Patients 17 and 38 were diagnosed at 89 and 33 months with *PAX2* genetic missense variant of c.122T>C (p.I41T) and c.254G>T (p.Gly85Val), respectively. Both showed initial SRNS; the former did not use immunosuppressants, whereas the latter was unresponsive to immunosuppressants. Currently, two patients with SNRS did not progress to CKD5 (**Tables 2 and 3**).

Patient 5, an 80-month-old girl, was admitted to the hospital with massive proteinuria owing to NS without any family history. She also had mental retardation and an atrial septal defect, with a karyotype of 46XY. A heterozygous pathogenic *WT1* variant c.1447+5G>A (splicing) was found. Physical examination revealed normal genitals, with no nephroblastoma (Wilms tumor), and the patient was diagnosed with Denys-Drash syndrome. As her kidney function rapidly declined to CKD5, kidney puncture was not performed, and she had been on

regular hemodialysis until the last follow-up. Patient 133, a 27-month-old boy with de novo nucleoporins variants, had a heterozygous nonsense variant of *NUP107* in exon 11c.935dup (p.Tyr312Ter). Rapid SRNS progression to CKD5 led to kidney transplantation without recurrence (**Tables 2 and 3**).

## SRNS prognosis

CKD at disease onset was observed in 18 (22.2%) patients after a median follow-up of 39 months (IQR 24, 65). When patients were stratified based on identified genetic variants, the group with identified genes had higher frequencies of CKD5 ($P = 0.006$), whereas the group with unidentified genes had a higher proportion of participants with normal eGFR ($P<0.001$; **Table 1**).

Kaplan–Meier curves and log-rank tests illustrated the progression rate to CKD in SRNS or infantile NS between the two groups. Participants with identified genetic variants were approximately ten-fold more likely to develop CKD (HR, 10.227; 95%CI, 3.505–29.838; $P<0.001$) than those in the unidentified genetic variant group at the last follow-up (**Fig 3**).

Univariate Cox regression analysis revealed that genetic variant, family history, extra-kidney manifestations, serum creatinine, hypertension, and response to immunosuppressants may be risk factors for CKD ($P<0.05$). Finally, we conducted multivariate Cox regression analysis and identified gene variant (HR, 5.110; 95%CI, 1.122–23.271; $P = 0.035$) as an independent predictor significantly associated with CKD (**S2 Table**). These results suggest that the gene variant group has a higher risk of CKD than the without gene variant group.

Furthermore, we conducted a subgroup analysis with four factors (including age, family history, extra-kidney manifestations, and hypertension) to explore whether the effect of gene variants on CKD varies with different population characteristics. **Fig 4** presents the results of the subgroup analysis. No significant differences were observed in kidney function in the subgroup analyses, with the possible exception of the family history level at baseline. The effect on kidney function in the identified variant group may be more pronounced among patients with a negative family history (HR: 15.325, 95%CI: 3.593–65.356) than those with a positive family history (HR: 1.656, 95%CI: 0.319–8.582; $P = 0.047$) for interaction. Nevertheless, these findings are exploratory because of a lack of adjustment for the multiplicity of hypothesis tests, and the research sample size is small.

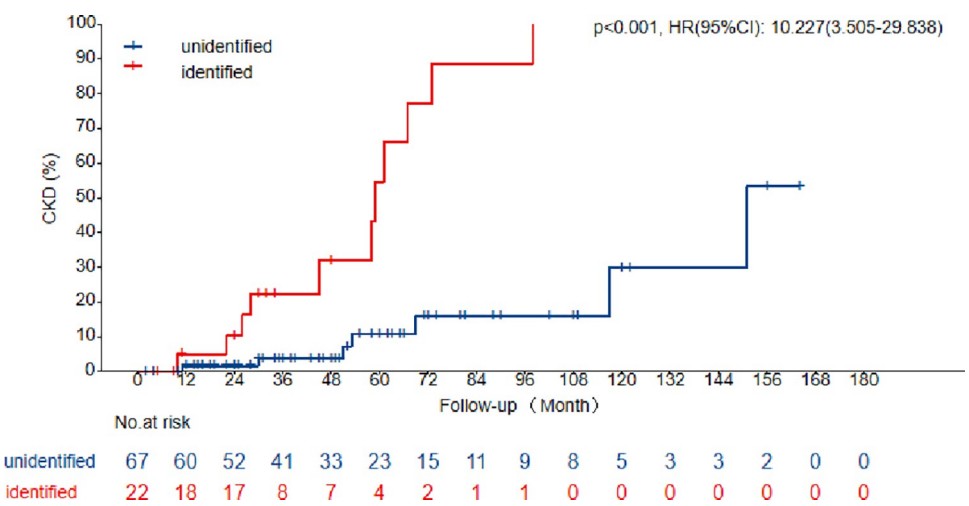

**Fig 3. Kaplan–Meier curves illustrate the progression to CKD in identified and unidentified genetic variants.** Endpoint at CKD5.

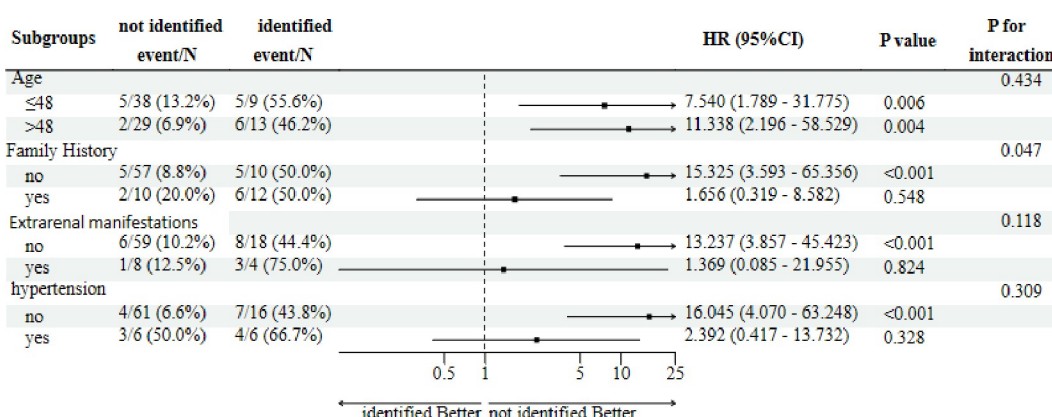

**Fig 4. Results of the subgroup analysis on gene mutation in CKD.**

## Discussion

In our cohort study, 89 patients with SRNS or infantile NS were recruited from five pediatric nephrology centers in Guangxi Zhuang Autonomous Region, with 22 (24.7%) cases identified with genetic variants. Compared with previous large international cohorts from Western European/North American studies [4–6,19] and Africa and Asia multicenter cohort studies [7,8,20,21], we observed similar variant frequencies. Nevertheless, some discrepancies were observed. First, contrary to the notion that "an early age of onset increases the likelihood of detection of causative variants" [5,7,22], we did not discover a higher variant rate in younger age groups. The mean age of the identified variant group was much older than the unidentified groups in our research. This difference may be attributed to the under-representation of infantile NS in our study (age of onset<3 months, 1 of 89; age 3–12 months, 7 of 89), along with the high infant mortality rates in our region owing to low income and a lack of medical resources. Additionally, the limitations of WES and insufficiency of relevant genetic studies should be acknowledged. WES method is suitable for the exon region of the gene, excluding the non-coding region variants such as the promoter region. Second, we detected three congenital nephrotic syndrome (CNS) gene variants, NPHS2 (five cases), NPHS1 (two cases), and WT1 (one case) in eight patients with a mean age of 52 months. In a study from Japan, NPHS1, LAMB2, and WT1 were reported as the most commonly detected genes in CNS [23], whereas in North America [4], the distribution of causative genes in CNS was reported as NPHS1, PLCE1, and NPHS2. Multicenter studies in other parts of China revealed different distributions of causative genes in CNS, including NPHS1, WT1, NPHS2, LAMB2, and COQ8B [7]. These variations highlight the differences in genetic detection rates and causative genes in CNS among different countries and districts worldwide.

Third, another striking finding in our study was the much higher incidence of *COL4A5* variants, accounting for 45.4% (10/22) of patients with monogenic SRNS and 11.2% (10/89) of all children with SRNS. This rate contrasts with the lower percentages reported in Western Europe (6.1%/1.6%) [6], North America (4.1%/1.0%) [4], Japan (2.8%/0.9%) [8], Korea (8.7%/3.8%) [21]. *COL4A5* encodes one of the six subunits of type IV collagen, a major structural component of basement membranes, and is the commonest cause of AS, occurring in 1 out of 2320 individuals [24]. A recent study by Sinha reported a high fraction of *COL4A* genetic variants among a cohort of children with SRNS from Eastern India [22], aligning with our results. It is intriguing that 90% of the ten patients with COL4A5 gene variants are of Han nationality, only 1 case is of other ethnic groups. Whether has an ethnic tendency remains uncertain, there

are no reports of COL4A5 gene variants in other regions with different ethnic groups in China till now. Relevant *COL4A3/A4/A5* genes are increasingly recognized to define the pathogenesis of familial FSGS or SRNS, expanding beyond the classic AS phenotype [25]. *COL4A* variants are also more readily identified in patients with isolated proteinuria phenotype or FSGS among monogenic nephropathy, especially in adults [4,26]. Compared with patients with SRNS and *COL4A* variants in our study, early reports in adults suggested the possibility of secondary FSGS, such as obesity, but this may be coincidental [27]. The question arises: what do *COL4A5* gene variants represent in our study––do they present an atypical form of AS or a distinct type of SRNS?

Among all patients with COL4A5 gene variants, 9 of 10 did not exhibit obvious symptoms of AS. Kidney pathology revealed FSGS or MCD, consistent with findings in other studies showing these conditions occurring in most children with SRNS and COL4A variants [19,22]. This finding can be attributed to an incomplete phenotype of COL4A5 gene variants. The high-frequency family history (70.0%), diverse changes observed in EM results(77.8%), and response to immunosuppressants support our inference. However, in the study by Sinha et al, 4 cases of 8 SRNS with COL4An genetic variations had reliable EM data, of which only 2 patients had glomerular basement membrane (GBM) changes [22]. It is widely recognized that as individuals age, they may experience kidney abnormalities, as well as ocular and auricular alterations. Consequently, within our cohort, it is acknowledge that some patients with a COL4A5 gene variant cannot entirely dismiss the possibility of AS and should undergo regular monitoring in the later stages. But, it is plausible to speculate that variants in the COL4An gene may have implications beyond the scope of AS disease. In the early stages of COL4A5 gene variant, the primary clinical symptoms may encompass not just hematuria, also with massive proteinuria. Previous reports have indicated that abnormal matrix-podocyte interactions, defective expression, or trafficking of the GBM matrix components could explain the relationship between *COL4A* variants and SRNS/FSGS [26]. Moreover, as early-stage disease is often clinically silent, the lack of clinical phenotype, pathological data, or discrepancies in genotype and phenotype could complicate the interpretation. Identifying genetic variants avoids exposure to immunosuppressive regimens, which are inevitably associated with an increased risk of infection and other drug side effects. Regardless, *COL4A5* gene variants accounted for a large proportion of gene variants in our cohort study. Combined with other previous reports, this finding is not accidental, especially in cases with a family history, which are more likely to be detected with *COL4A* variants. Furthermore, patient 62 adds to the growing evidence that patients carrying variants in an SRNS/FSGS gene and *COL4A* have increased disease severity [28], supporting the idea that variants in different genes that converge in the glomerular filtration barrier may accelerate disease progression.

The results in our cohort were comparable to those of other countries regarding the response of SRNS to CNIs, which are the first-line immunosuppressive treatment for patients with SRNS [29]. The percentages of complete remission in the gene variant and non-variant groups were 9.1% and 71.6%, respectively. A recent retrospective study on 141 patients identified with pathogenic or probably pathogenic variants in monogenic causes of SRNS revealed that 27.6% responded to CNIs, of which 75.0% maintained stable kidney function. This is the first evidence that CNIs can be effective in children with genetic SRNS, increasing kidney survival and reducing the need for kidney replacement therapy [30]. This efficacy may be related to the fact that CNIs can maintain kidney function by stabilizing synaptopodin in podocytes [31]. Other reports present varying results on treatment efficacy in patients with monogenic SRNS [32,33]. Elshafey et al. also revealed that *COL4A* variants can achieve spontaneous remission in Egyptian children [34]. Conversely, a study by Di et al. supported that early long-term treatment with angiotensin-converting enzyme inhibitors/angiotensin receptor blockers can

provide better benefits in some patients with *COL4A5* gene variants [35]. Therefore, treating monogenic SRNS remains challenging. However, these studies indicate that CNIs may not be useless in treating monogenic SRNS, emphasizing the need for more prospective randomized trials with longer follow-ups to verify these findings.

The probability of progressing to CKD5 in children with SRNS at 5 years after diagnosis ranged from 8% to 35% [36]. Reports on the prognosis of genetic SRNS are currently limited. Frameshift variants and splicing site alterations often result in a more severe phenotype than missense variants [37]. Because of the small number of pathogenic gene variants detected in our study, we didn't perform statistical analysis of the relationship between variant types and prognosis. However, we concluded that in our cohort, the risk of SRNS progressing to CKD was higher in the identified group than in the unidentified group. Moreover, gene variant may be an independent predictor of SRNS progression to CKD. However, the small sample size and discrepancies in treating each genetic variant hinder our ability to identify prognostic differences between different gene variants.

Our study is limited by its retrospective design and the small number of patients, hindering unequivocal statistical comparisons. Moreover, these studies typically originate from genetic laboratories and may be subject to referral bias, as patients with a young age of onset, a positive family history, or syndrome features may be more likely to undergo genetic testing. Additionally, the short follow-up period limits our accurate assessment of the prognosis.

In conclusion, we observed a genetic yield of 24.7% among this multicenter cohort of children with SRNS who underwent WES. The detection rate of gene variants in late SRNS is low. Furthermore, we confirmed that *COL4A5* variants are the most common among children with SRNS in Guangxi, suggesting that *COL4A5* may be a crucial predictive genetic factor in SRNS in this region. Moreover, gene variant may be an independent predictor of SRNS progression to CKD.

## Supporting information

**S1 Table. Extra-kidney manifestations of 12 SRNS.**
(PDF)

**S2 Table. Univariate Cox regression analysis with 2 group progression to CKD.**
(PDF)

**S1 Fig. Extra-kidney manifestations of 12 SRNS, Mental retardation is most frequent(5/ 12),and follow by Atrial septal defect(4/12),only one had hearing loss at the initiate of disease.**
(PDF)

**S2 Fig. Univariate Cox regression analysis with 2 group progression to CKD.**
(PDF)

**S1 Raw data.**
(XLS)

## Author Contributions

**Conceptualization:** Sheng Li, Miaoyue Hu, Yuanhan Qin.

**Data curation:** Sheng Li, Weifang Huang, Fengying Lei, Yunguang Liu, Zengpo Huang, Yongqiu Meng, Wenjing Liu, Xianqiang Lei, Yanfang Dong, Zihui Lin, Chunlin Huang, Rihong Zhao.

**Formal analysis:** Sheng Li, Chao He, Yu Sun.

**Methodology:** Sheng Li, Yuanhan Qin.

**Supervision:** Yuanhan Qin.

**Writing – original draft:** Sheng Li, Yuanhan Qin.

**Writing – review & editing:** Sheng Li, Yuanhan Qin.

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
