## [Decision Letter · Decision Letter 0]

19 Feb 2024

PONE-D-23-42717A multicenter study investigating the genetic analysis of childhood steroid-resistant nephrotic syndrome: mutations in COL4A5  may not be coincidentalPLOS ONE

Dear Dr. Qin,

Thank you for submitting your manuscript to PLOS ONE. After careful consideration, we feel that it has merit but does not fully meet PLOS ONE’s publication criteria as it currently stands. Therefore, we invite you to submit a revised version of the manuscript that addresses the points raised during the review process.

If applicable, we recommend that you deposit your laboratory protocols in protocols.io to enhance the reproducibility of your results. Protocols.io assigns your protocol its own identifier (DOI) so that it can be cited independently in the future. For instructions, see: https://journals.plos.org/plosone/s/submission-guidelines#loc-laboratory-protocols. Additionally, PLOS ONE offers an option for publishing peer-reviewed Lab Protocol articles, which describe protocols hosted on protocols.io. Read more information on sharing protocols at https://plos.org/protocols?utm_medium=editorial-email&utm_source=authorletters&utm_campaign=protocols.

We look forward to receiving your revised manuscript.

Kind regards,

Rami M. Elshazli, Ph.D

Academic Editor

PLOS ONE

Journal Requirements:

"All the authors have declared no competing interests."

Reviewers' comments:

Reviewer's Responses to Questions

**Comments to the Author**

1. Is the manuscript technically sound, and do the data support the conclusions?

Reviewer #1: Yes

Reviewer #2: Yes

Reviewer #3: Partly

2. Has the statistical analysis been performed appropriately and rigorously? 

Reviewer #1: I Don't Know

Reviewer #2: Yes

Reviewer #3: Yes

3. Have the authors made all data underlying the findings in their manuscript fully available?

Reviewer #1: Yes

Reviewer #2: Yes

Reviewer #3: Yes

4. Is the manuscript presented in an intelligible fashion and written in standard English?

Reviewer #1: Yes

Reviewer #2: Yes

Reviewer #3: Yes

5. Review Comments to the Author

Reviewer #1: 1. The definition of early onset is unusual. Suggest change to congenital or infantile nephrotic syndrome.

2. For consistency, give the percentage to one decimal space.

3. What do the authors mean by resistant style? Is this steroid resistance?

4. Do the authors mean patients with INS who are steroid-resistant?

5. For the table, a more detailed explanation as a legend or the heading must be stated as to what “not

identified” or “identified” refers to.

6. For neonates, the best formula for eGFR using serum creatinine is the Brion et al. formula:

(eGFR = k × Ht/SCr, k = 0.33 [preterm], k = 0.45 [term infants]), which is considered the most appropriate f.

for estimating GFR in the neonatal population. (Ref J Pediatr Pharmacol Ther. 2018 Nov-Dec; 23(6): 424–431.

doi: 10.5863/1551-6776-23.6.424). This is better than the modified Swartz formula.

7. Do the authors mean “chronic kidney failure (CKF)” or progression of CKD.

8. Reference 3: References incorrectly cited. Please correct the citation.

9. The definition of early onset is unusual. Suggest a change to congenital or infantile nephrotic syndrome.

Reviewer #2: Over all, the manuscript entitled “A multicenter study investigating the genetic analysis of childhood steroid-resistant nephrotic syndrome: mutations in COL4A5 may not be coincidental” is of interest and Adding some new data in this field.

Reviewer #3: Dear Sir: Editor-in-Chief

Thank you for your kind invitation to review this article entitled" A multicenter study investigating the genetic analysis of childhood steroid-resistant nephrotic syndrome: mutations in COL4A5 may not be coincidental.". I found the article very interesting, coupled with the era of genetic understanding of pediatric renal diseases; it is well written with clear language, but I have some comments that the authors should declare.

Major comments:

1. You mentioned that the study aimed to discuss the pathogenic hereditary factors of SRNS. At the same time, you included cases with congenital nephrosis ( age ≤ 3 months) that can not be classified as SRNS, as those cases are not eligible for immunosuppressive therapy. Please exclude this group from the study.

2. Why were only 66 patients out of 89 underwent renal biopsy despite all being considered SRNS. The guidelines mandate renal biopsy in all cases with SRNS , so please declare why biopsy was not done in 23 patients.

3. Why did 13 of your patients not receive immunosuppressives despite being SRNS?

4. How can you explain the high incidence of MCD in your cohort while previous reports from India stated that 45% of SRNS are FSGS?

5. What was the indication for WES in children with late steroid resistance with MCD and normal kidney function in the light of absent consanguinity and negative family history? This group of patients are not likely to have monogenic SRNS, as your findings prove.

6. Regarding the family history, I think it essential to declare FH of hematuria or extra-renal manifestations as SNHL, which suggests Alport syndrome, especially in cases with COL4A5 mutations.

7. You reported nephritic clinical style in 13/22 cases, but you didn't mention the association between this clinical style and the reported genotypes.

8. It is well-known that COL4A5 is associated with Alport syndrome and 2ry NS. Out of the 10 reported cases, most of them were from Han ethnicity, age > 6 years at onset, family history was reported in 7 patients, all of them were resistant to immunosuppressive, and sensory neural deafness was already reported in one of them. How did you exclude Aloprt syndrome in these cases, considering that the diffuse thinning of GBM by EM supports the diagnosis of Alport syndrome? Many previous studies, including some you cited, have stated that FSGS can be either a phenocopy or late pathology of AS. For example, In the study by ElShafey et al (reference 36), they reported that the father of the only case with isolated NS and COL4A5 mutations had CKD and sensory-neural deafness. Also, in the study by Gast et al. (reference 28), only 2/8 patients with COL4A mutations had SNHL before they were 40 years of age. Laking of the extra-renal manifestations of AS in your group can't exclude it as they may appear later in adulthood. So, I believe you can't conclude that cases of FSGS and COL4A5 mutations are primary FSGS rather than AS with phenocopied FSGS. It is better to figure out that AS is a common undiagnosed cause of SRNS in India, especially the Han ethnicity in the discussion section

9. Minor comments

In Table 1 ; the BP was significantly higher in the identified group, which is logical as they have significantly older ages, while the BP percentile was comparable across the 2 groups which is more important than the rough values of BP

In the genotype=phenotype correlations section replace men and women with, males and females.

6. PLOS authors have the option to publish the peer review history of their article (what does this mean?). If published, this will include your full peer review and any attached files.

Reviewer #1: No

Reviewer #2: **Yes: **Kinnari N Mistry, Ph.D, Associate Professor , ASHOK & RITA PATEL INSTITUTE OF INTEGRATED STUDY & RESEARCH IN BIOTECHNOLOGY AND ALLIED SCIENCES (ARIBAS) - A Constituent College of CVM University, Vallabh Vidyanagar, Gujarat, India

Reviewer #3: No

---

## [Author Response · Author response to Decision Letter 0]

5 Mar 2024

Dear Editors and Reviewers:

Thank you very much for your comments and recognition on the manuscript entitled “A multicenter study investigating the genetic analysis of childhood steroid-resistant nephrotic syndrome: mutations in COL4A5 may not be coincidental”. 

Based on your comment and request, we have made extensive modification on the original manuscript (including ‘Funding Information’, ethics statement in the ‘Methods’ section). Here, we attached revised manuscript in doc for your approval. A document answering every question from the referees was also summarized and enclosed. Should you have any questions, please contact us without hesitate.

Those comments are all valuable and very helpful for revising and improving our paper, as well as the important guiding significance to our researches. We have studied comments carefully and have made correction which we hope meet with approval. Revised portion are marked in highlighted in the paper. The main corrections in the paper and the responds to the reviewer’s comments are as following:

Reviewer #1: 

1.The definition of early onset is unusual. Suggest change to congenital or infantile nephrotic syndrome.

Respond: Thanks for your suggestion, we have modified early-onset nephrotic syndrome to infantile nephrotic syndrome. Line25,97,169,209,291,327,338.

2.For consistency, give the percentage to one decimal space.

Respond: Thank you for your careful reminder, we have revised the percentage description in the article to one decimal place. The place of modification in Table 1.

3.What do the authors mean by resistant style? Is this steroid resistance?

Respond: Yes, Thank you for your suggestion. We are very sorry for our negligence. We have modified resistant style to steroid resistance style as you suggested. Table 1.

4.Do the authors mean patients with INS who are steroid-resistant?

Respond: Thanks for your reminder. Yes, INS refers to idiopathic nephrotic syndrome, the first-line medication of INS is steroid, which can be divided into steroid-sensitive and steroid-resistant type according to the response to steroid, and the latter is included in our study. 

5.For the table, a more detailed explanation as a legend or the heading must be stated as to what “not identified” or “identified” refers to.

Respond: Thank you for your suggestion,we have modified and explain.Line 137,138

6.For neonates, the best formula for eGFR using serum creatinine is the Brion et al. formula:(eGFR = k × Ht/SCr, k = 0.33 preterm, k = 0.45 term infants), which is considered the most appropriate f.for estimating GFR in the neonatal population. (Ref J Pediatr Pharmacol Ther. 2018 Nov-Dec; 23(6): 424–431.doi: 10.5863/1551-6776-23.6.424). This is better than the modified Swartz formula.

Respond: Thank you for your suggestion. Apologize for the misunderstanding caused by our unclear statement. Actually, In this study the age of the child refers to the age at the time of onset, while the calculated value of eGFR and Scr corresponds to the evaluation of renal function at the last follow-up or hospitalization. And we have re-commented on line 118. Therefore, although we included a 1month infantile (age of onset), the eGFR record was his last follow-up.Thank you again.

7.Do the authors mean “chronic kidney failure (CKF)” or progression of CKD.

Respond: Thank you for your reminder. Chronic renal failure(CKF) belongs to the category of chronic kidney disease (CKD), and the scope of CKD is more extensive. CKD in this article refers to eGFR less than 90 mL/min/1.73 m2. We generally think that CKD5 is equivalent to end-stage kidney disease. In this study, eGFR was determined based on Scr at the last follow-up to evaluate renal function (normal renal function, CKD grade).

8.Reference 3: References incorrectly cited. Please correct the citation.

Respond: Thank you for your careful reminder. We are very sorry for our negligence. The place of modification,Line487-489.

9.The definition of early onset is unusual. Suggest a change to congenital or infantile nephrotic syndrome.

Respond: Consistent with the answer to comment 1. Thanks for your suggestion.

Reviewer #3: 

Major comments:

1.You mentioned that the study aimed to discuss the pathogenic hereditary factors of SRNS. At the same time, you included cases with congenital nephrosis ( age ≤ 3 months) that can not be classified as SRNS, as those cases are not eligible for immunosuppressive therapy. Please exclude this group from the study.

Respond: Thank you for your suggestion. Yes, among the SRNS cases included in this study, there was a child with congenital nephrotic syndrome (CNS) less than 3 months old who was not treated with steroid or immunosuppressants immediately after diagnosis. Actively improve the gene detection: TBX18,c.1100-5T＞C (p.?), ACMG classification is uncertain, the mutation gene is related to congenital anomalies of the kidney and urinary tract (CAKUT). Due to the age did not performed the renal pathological. In the later stage, the urine protein did not turn negative after a sufficient course of steroid treatment, and became negative after the addition of TAC. As of the time of submission, the child was 2 months off 4 years old with normal renal function. So,we included this case.

2.Why were only 66 patients out of 89 underwent renal biopsy despite all being considered SRNS. The guidelines mandate renal biopsy in all cases with SRNS , so please declare why biopsy was not done in 23 patients.

Respond: Thank you for your reminder. Yes, renal pathological examination should be performed in all cases in our cohort study as recommended by the guidelines, but in fact 33 children didn’t. These included 23 cases under 5 years old and 10 cases older than 5 years old. Reasons for not complete the renal pathology were the age of the patients, general anesthesia and family members' refusal to perform invasive operation.

3.Why did 13 of your patients not receive immunosuppressives despite being SRNS?

Respond: Thank you for your careful reminder.As shown in the following Table 1.13 SRNS patients did not receive immunosuppressants, including 6 with genetic variants. Four of them were not given because of the continuous decrease of renal function, four patients refused to use it and the other five patients were received RTX.

Table 1 Details of 13 SRNS without receive immunosuppressant 

Patients ID Age(Month) Gene results Reasons of did not use immunosuppressives

2 168 COL4A5 Refuse

17 89 PAX2 Decrease of renal function

5 80 WT1 Decrease of renal function

48 20 Negative Decrease of renal function

70 6 Negative Refuse

78 72 Negative Used of RTX

87 48 LMX1B Decrease of renal function

105 132 COL4A5 Refuse

113 19 Negative Used of RTX

266 24 Negative Used of RTX

270 15 Negative Used of RTX

278 37 Negative Used of RTX

281 89 COL4A5 Refuse

RTX:Rituximab

4.How can you explain the high incidence of MCD in your cohort while previous reports from India stated that 45% of SRNS are FSGS?

Respond: Thanks you for your question.Among the 56 children with SRNS who underwent renal pathological in this study, MCD and FSGS accounted for 51.8% and 32.1% respectively, which were different from the previous reports. Possible reasons, firstly, thirty-three cases in this study didn’t complete the kidney biopsy; Secondly, multiple single-center studies have reported that the detection frequency of FSGS varies in different ethnic backgrounds [1-4]. Some researches shown that the incidence of FSGS increases with age. In this study, the median age of SRNS in 56 cases who performed renal biopsy is 73.5 months (4.00, 110.25), which may be the reason for the low detection rate of FSGS. Thirdly, it has also been reported that MCD and FSGS may be different pathological changes in the development of the same disease and MCD may develop to FSGS with the progress of the disease [5].

5.What was the indication for WES in children with late steroid resistance with MCD and normal kidney function in the light of absent consanguinity and negative family history? This group of patients are not likely to have monogenic SRNS, as your findings prove.

Respond: Thank you for your good question. Yes, according to 2020 IPNA's recommendations for the diagnosis and treatment of steroid-resistant nephrotic syndrome in children, genetic testing for late-onset steroid-resistant nephrotic syndrome is not recommended (grade C, moderately recommendation) [6]. At present, gene mutations account for only 20-30% of all SRNS cases with single gene mutations, indicates that SRNS has genetic heterogeneity, finding and clarifying more pathogenic gene mutations is the purpose of our multi-center study. Our results are also as pointed out by the professor that the positive rate of gene mutation in children with late steroid resistant nephrotic syndrome is low, but it is recommended to improve gene detection in children with family history and extra-renal phenotype.

6.Regarding the family history, I think it essential to declare FH of hematuria or extra-renal manifestations as SNHL, which suggests Alport syndrome, especially in cases with COL4A5 mutations.

Respond: Thanks very much for your advice. We reorganized the family history of children with COL4A5 pathogenic gene mutation. As shown in Table 2, there were 10 cases of SRNS complicated with COL4A5 and 7 cases with family history.Patients2’s mother had hematuria, Patients281’s father and brother had hematuria, Patients107 was adopted by their parents. In addition, none of the families had a history of hematuria.

Table 2 Family history and extra-renal manifestations of SRNS with COL4A5 gene mutation

Patients ID Age(Month) FH(Family history and hematuria ) Extra-renal manifestations 

2 168 Brother CKD5 (cause unknown),mother has hematuria -

61 147 Mother has a history of nephritis SNHL

128 115 Grandmother has a history of nephritis -

129 115 Grandmother has a history of nephritis -

16 124 Grandfather has a history of nephritis -

221 94 Aunt has a history of nephritis -

281 89 Grandfather CKD5,father and brother has hematuria -

43 33 Negative -

107 72 Adopted child -

105 132 Negative -

SNHL:Sensorineural hearing loss 

7.You reported nephritic clinical style in 13/22 cases, but you didn't mention the association between this clinical style and the reported genotypes.

Respond: Very appreciate for the professor's reminder. In the early years of China, it is customary to classify children with INS into simple type and nephritis type according to clinical classification, but there is no such classification abroad. With the extensive development of renal biopsy and gene detection, the clinical typing diagnosis of INS has been gradually replaced. Therefore, this article has removed this classification from Table 1.

8.It is well-known that COL4A5 is associated with Alport syndrome and 2ry NS. Out of the 10 reported cases, most of them were from Han ethnicity, age > 6 years at onset, family history was reported in 7 patients, all of them were resistant to immunosuppressive, and sensory neural deafness was already reported in one of them. How did you exclude Aloprt syndrome in these cases, considering that the diffuse thinning of GBM by EM supports the diagnosis of Alport syndrome? Many previous studies, including some you cited, have stated that FSGS can be either a phenocopy or late pathology of AS. For example, In the study by ElShafey et al (reference 36), they reported that the father of the only case with isolated NS and COL4A5 mutations had CKD and sensory-neural deafness. Also, in the study by Gast et al. (reference 28), only 2/8 patients with COL4A mutations had SNHL before they were 40 years of age. Laking of the extra-renal manifestations of AS in your group can't exclude it as they may appear later in adulthood. So, I believe you can't conclude that cases of FSGS and COL4A5 mutations are primary FSGS rather than AS with phenocopied FSGS. It is better to figure out that AS is a common undiagnosed cause of SRNS in India, especially the Han ethnicity in the discussion section.

Respond: Thanks very much for professor's question. Indeed, the diagnosis of these ten patients with COL4A5 gene mutation is controversial. As is known to all, the diagnosis of SRNS is based on clinical diagnosis, and the pathological results have no obvious specificity,about 30% of the children progressed to CKD5. And the diagnosis of AS mainly depends on renal pathology (especially the results of electron microscope),the abnormality of type Ⅳ collagen in basement membrane is the pathogenesis of the disease, and the prognosis of female patients is better than that of men, gene detection is considered as an important strategy for the diagnosis of this disease. First of all, the clinical analysis of 10 children with massive albuminuria as the first manifestation of edema, hypoproteinemia. Male:female = 8:2. Among the two female children, case 2 had family history and mother had hematuria history, but the renal pathological examination was not perfect. Patient 43 had no family history, and the pathological results suggested that there were uneven changes in the thickness of the basement membrane, so the above two patients could not be clearly diagnosed as INS or atypical AS before the results of gene test.

As below, we will analyze the clinical features, pathological diagnosis, gene test results and prognosis of these 10 cases one by one. Firstly, all the 10 patients with massive albuminuria as the first manifestation of edema, hypoproteinemia. Male :female = 8:2. Seven cases had family histories. Hearing loss was found in one patient. None of them had ocular changes. All of genetic mutation mode is XLD. 

Of the two female patients, patient 2 had family history and mother has hematuria, without extra-renal manifestations, but didn’t perfect renal pathological. Patent 43 had no family history, and the pathological results showed that there was uneven thickness of the GBM. The mutation types of the two cases were splicing and missense mutation, respectively. For the treatment, before the gene test results were given sufficient steroid, but the urine protein did not turn negative, the former was not given immunosuppressant therapy, the latter was unresponsive to immunosuppressants. However, these two children still had urinary protein, they had no vision and hearing loss, and their renal function remained normal. Therefore, although the above-mentioned children were treated with SRNS, AS could not be excluded, and renal function, hearing and fundus examination should be monitored regularly.

Among the other eight male children, patient 61 had family history and had hearing loss at the onset of the disease, but the pathological results showed that it was FSGS, and there were no obvious characteristic changes in the GBM under EM. Patient128,129 were twin brothers and great-grandmother had a history of CKD5,EM results showed slight thinning of basement membrane and frameshift mutation. The grandfather of patient 16 had a history of glomerulonephritis, EM showed partial tearing of the GBM. The aunt of case 221 had a history of glomerulonephritis,the pathological results showed MN under light microscope and immunofluorescence, and AS was considered by EM, the type of gene mutation was frameshift mutation this patient was ineffective to immunosuppressant therapy and eventually died after CKD5. The grandfather of case 281 had a history of CKD5, and both father and brother had hematuria, did not use immunosuppressant, pathological results showed MCD, and the EM also no obvious change in the GBM, gene detection showed that there were missense mutations at two sites, but the renal function was still normal. Patient 105,107 had no family history,the pathological results showed that the GBM was diffusely thinned, the former was not treated with immunosuppressant, and the latter was ineffective. Both of them were in CKD2 stage.In summary, except for renal manifestations (hearing loss, ocular changes), combined with family history and renal pathological examination, 6 of 10 children need long-term follow-up to rule out AS.In addition, according to your suggestion, we have

---

## [Decision Letter · Decision Letter 1]

1 Apr 2024

PONE-D-23-42717R1A multicenter study investigating the genetic analysis of childhood steroid-resistant nephrotic syndrome: mutations in COL4A5  may not be coincidentalPLOS ONE

Dear Dr. Qin,

Thank you for submitting your manuscript to PLOS ONE. After careful consideration, we feel that it has merit but does not fully meet PLOS ONE’s publication criteria as it currently stands. Therefore, we invite you to submit a revised version of the manuscript that addresses the points raised during the review process.

We look forward to receiving your revised manuscript.

Kind regards,

Rami M. Elshazli, Ph.D

Academic Editor

PLOS ONE

Journal Requirements:

Reviewers' comments:

Reviewer's Responses to Questions

**Comments to the Author**

Reviewer #1: All comments have been addressed

Reviewer #2: All comments have been addressed

2. Is the manuscript technically sound, and do the data support the conclusions?

Reviewer #1: Yes

Reviewer #2: Partly

3. Has the statistical analysis been performed appropriately and rigorously? 

Reviewer #1: I Don't Know

Reviewer #2: Yes

4. Have the authors made all data underlying the findings in their manuscript fully available?

Reviewer #1: Yes

Reviewer #2: Yes

5. Is the manuscript presented in an intelligible fashion and written in standard English?

Reviewer #1: Yes

Reviewer #2: Yes

6. Review Comments to the Author

Reviewer #1: The manuscript has been greatly improved but needs some minor revisions. The word "renal" should be changed to "kidney".

The sentences NS (massive proteinuria) probably an early phenotype of

COL4A5 gene mutations" (line 387) needs to be explained.

Change the word "mutations" to "variants".

Reviewer #2: A multicenter study investigating the genetic analysis of childhood steroid-resistant

nephrotic syndrome: mutations in COL4A5 may not be coincidental

Submitted by Authors may be accepted for publication in present form.

7. PLOS authors have the option to publish the peer review history of their article (what does this mean?). If published, this will include your full peer review and any attached files.

Reviewer #1: **Yes: **Rajendra Bhimma

Reviewer #2: **Yes: **Dr. Kinnari Nitin Mistry

---

## [Author Response · Author response to Decision Letter 1]

5 Apr 2024

Dear Editors and Reviewers:

Thank you very much for your comments and recognition on the manuscript entitled “A multicenter study investigating the genetic analysis of childhood steroid-resistant nephrotic syndrome: variants in COL4A5 may not be coincidental”. 

Based on your comment and request, we have made minor modification on the original manuscript, the funding sources were disclosed in the cover letter and their roles were specified in the manuscript. Any funding-related details have been removed from our manuscript.Here, we attached revised manuscript in doc for your approval. A document answering every question from the referees was also summarized and enclosed. Should you have any questions, please contact us without hesitate.

Those comments are all valuable and very helpful for revising and improving our paper, as well as the important guiding significance to our researches. We have studied comments carefully and have made correction which we hope meet with approval. The main corrections in the paper and the responds to the reviewer’s comments are as following:

Reviewer #1: 

1.The word “renal” should be changed to “kidney”.

Respond: Thanks for your careful reminder, we have modified the word “renal” to “kidney”.

2.The sentences NS (massive proteinuria) probably an early phenotype of COL4A5 gene mutations (line 387) needs to be explained.

Respond: Thanks for your suggestion, the sentences “NS (massive proteinuria) probably an early phenotype of COL4A5 gene mutations” have explained to “In the early stages of COL4A5 gene variant, the primary clinical symptoms may encompass not just hematuria, also with massive proteinuria.”Line388-390.

3.Change the word “mutations” to “variants”.

Respond: Thanks for your suggestion, we have modified the word “mutations” to “variants”.

We tried our best to improve the manuscript and made some changes in the manuscript.  These changes will not influence the content and framework of the paper. And here we did not list all the changes but marked in highlight in revised paper. We appreciate for Editors/Reviewers’ warm work earnestly, and hope that the correction will meet with approval. 

On behalf of my co-authors, we would like to expression our great appreciation to the editors and reviewers.

Yours sincerely,

Dr. Yuanhan,Qin,

Department of Pediatrics, 

The First Hospital of Guangxi Medical University, 

Nanning, Guangxi,China, 

Tel: 86-13607819988

E-mail address: 

qinyuanhan2022@163.com

---

## [Decision Letter · Decision Letter 2]

21 May 2024

A multicenter study investigating the genetic analysis of childhood steroid-resistant nephrotic syndrome: variants in COL4A5  may not be coincidental

PONE-D-23-42717R2

Dear Dr. Qin,

We’re pleased to inform you that your manuscript has been judged scientifically suitable for publication and will be formally accepted for publication once it meets all outstanding technical requirements.

Kind regards,

Rami M. Elshazli, Ph.D

Academic Editor

PLOS ONE

Additional Editor Comments (optional):

Reviewers' comments:

Reviewer's Responses to Questions

**Comments to the Author**

1. If the authors have adequately addressed your comments raised in a previous round of review and you feel that this manuscript is now acceptable for publication, you may indicate that here to bypass the “Comments to the Author” section, enter your conflict of interest statement in the “Confidential to Editor” section, and submit your "Accept" recommendation.

Reviewer #3: (No Response)

2. Is the manuscript technically sound, and do the data support the conclusions?

Reviewer #3: Yes

3. Has the statistical analysis been performed appropriately and rigorously? 

Reviewer #3: Yes

4. Have the authors made all data underlying the findings in their manuscript fully available?

Reviewer #3: Yes

5. Is the manuscript presented in an intelligible fashion and written in standard English?

Reviewer #3: Yes

6. Review Comments to the Author

Reviewer #3: (No Response)

7. PLOS authors have the option to publish the peer review history of their article (what does this mean?). If published, this will include your full peer review and any attached files.

Reviewer #3: **Yes: **Heba M Ahmed

---

## [Editor Report · Acceptance letter]

24 May 2024

PONE-D-23-42717R2 

PLOS ONE

Dear Dr. Qin, 

I'm pleased to inform you that your manuscript has been deemed suitable for publication in PLOS ONE. Congratulations! Your manuscript is now being handed over to our production team.

Kind regards, 

on behalf of

Prof. Dr. Rami M. Elshazli 

Academic Editor

PLOS ONE